# A Rapid Prototyped Thermal Mass Flowmeter

**DOI:** 10.3390/s21165373

**Published:** 2021-08-09

**Authors:** Borut Pečar, Danilo Vrtačnik, Matic Pavlin, Matej Možek

**Affiliations:** Laboratory of Microsensor Structures and Electronics, Faculty of Electrical Engineering, University of Ljubljana, Tržaška 25, SI-1000 Ljubljana, Slovenia; danilo.vrtacnik@fe.uni-lj.si (D.V.); matic.pavlin@fe.uni-lj.si (M.P.); matej.mozek@fe.uni-lj.si (M.M.)

**Keywords:** thermal mass flowmeter, rapid prototyping, COMSOL, numerical simulation, calorimetric, hot wire

## Abstract

An innovative rapid prototyping technique for embedding microcomponents in PDMS replicas was developed and applied on a thermal mass flowmeter for closed loop micropump flowrate control. Crucial flowmeter design and thermal parameters were investigated with a 3-D fully coupled electro-thermal-fluidic model which was built in Comsol Multiphysics 5.2. The flowmeter was characterized for three distinct measuring configurations. For precise low flowrate applications, a sensor-heater-sensor flowmeter configuration with a constant heater temperature was found to be the most appropriate yielding the measuring range of 0 to 90 µL·min^−1^ and the sensitivity of 1.3 °C·µL^−1^·min in the lower flowrate range of 0 to 40 µL·min^−1^.

## 1. Introduction

Typically, a thermal mass flowmeter comprises a heating element which creates a local temperature increase and sensing elements that aim to measure the distortion in the temperature profile along the channel as induced by the fluid flow [1]. Two main principles of operation are a hot-wire and a calorimetric one. Several Si-based thermal flow sensors are currently available for measuring gas and liquid flowrates [2,3,4]. To enhance device sensitivity and to improve corresponding response time, heat dissipation to the substrate should be minimized [5]. Due to high thermal conductivity of a silicon substrate, various schemes are implemented in order to achieve a thermal isolation of sensing elements, such as free-standing structures, vacuum cavities and formation of a porous silicon or thin layer silicon nitride membranes [5]. However, most of these structures rely on advanced MEMS fabrication processes which require expensive facilities and skilled personnel. To provide a highly effective thermal isolation of heating/sensing elements without the need for the abovementioned thermal insulation approaches and additional MEMS fabrication processes, attempts were made to substitute a silicon substrate with organic materials such as SU8 [5], polyimide [6,7], Kapton^®^ [8] and parylene [9]. A major advantage arising from adoption of organic materials for a flowmeter substrate is their low thermal conductivity [5]. Although an employment of these organic materials simplified a flowmeter fabrication process, these materials are still not suitable for fabrication of microfluidic devices and systems in a rapid, inexpensive manner as demanded by current trends in microfluidic systems [10]. There is a growing need for simple, fast and low-cost rapid prototyping methods that can significantly speed up evaluation of proof-of-concept designs at an initial stage of laboratory research [11].

On the other hand, a polydimethylsiloxane (PDMS) polymer is an attractive material for rapid prototyping of microfluidic systems in laboratory environment [12]. It is tough, optically transparent, amenable to fabrication using a number of procedures, and relatively inexpensive [13]. It is non-polar, impermeable to aqueous solutions, well-suited for working with aqueous, biological samples, etc. [14]. As a result, numerous simple to employ, fast and cost effective processes for rapid prototyping of PDMS-based microfluidic devices have been developed and demonstrated [15,16,17]. Since a PDMS has the low thermal conductivity value of 0.15 Wm^−1^K^−1^ [18] it is a promising material for thermal insulation between heater/sensor elements in a flowmeter application.

Here, we present a novel rapid prototyping technique for embedding discrete components into PDMS replicas. Micro components are aligned onto a master and poured with liquid PDMS. After polymerization, a PDMS replica with embedded components is removed from the master being readily available for further process steps such as bonding with other layers etc. This novel technique was demonstrated on a thermal mass flowmeter integrated by a custom designed micropump (the micropump is not the subject of this work) on the same chip, where three commercially available Resistance Temperature Detector (RTD) Pt100 elements were embedded in the PDMS elastomer, as shown in Figure 1 and employed as heating and sensing elements. The proposed device is suitable for implementation in microfluidic systems which require flow control of reagents, but need to be fabricated in a rapid and cost efficient way and do not necessarily require high accuracy and fast response time of conventional MEMS micro flowmeters. Moreover, the device is suitable for integration on a common microfluidic platform with other PDMS fluidic components, such as micropumps, microvalves, reservoirs or micromixers.

## 2. Design

Figure 1 shows an exploded view drawing of the microfluidic chip comprising the piezoelectric single actuator peristaltic micropump [19] and the micro flowmeter represented by three embedded heating/sensing elements positioned downstream of the microchannel.

A microfluidic chip comprises the sandwich of three layers (glass–PDMS–glass). A PDMS replica forms a micropump chamber and a fluidic microchannel. It is bonded on its flat side to a thick glass substrate and on the opposite side with a thin glass membrane. One inlet and one outlet fluid port are pre-drilled through the glass substrate that supply and drain the fluid into and out of the chip. Additionally, two through-holes are punched into the elastomer, one into the center of the micropump chamber and the other one at the end of the channel. A piezoelectric actuator is positioned in the axis of the micropump chamber, coupled rigidly to the micropump membrane. A single actuator peristaltic micropump operates on a principle of active sequential expansion (opening) and compression (closing) of an inlet central port and an outlet channel [19].

By employing the innovative rapid prototyping technique (see Section 4), three heating/sensing elements are embedded in the elastomer along the path of the microchannel of which each one can be used as a heating element, a sensing element or a combination of both. Standard, commercially available platinum resistance elements Pt100 (DM-301, Labfacility Ltd., Southern Cross Industrial Estate, Shripney Road, Bognor Regis, UK) were chosen as the heating/sensing elements, primarily due to their ceramic base having a one flat side, what make them suitable for embedding.

The fluidic connections are realized with hollow PDMS cylinders which are bonded to the glass substrate. The polymer tubing is inserted in the cylinder’s cores.

Dimensions of the microfluidic chip were set as follows: The glass substrate, the PDMS elastomer layer and the glass membrane are rectangular in shape with the dimensions (L × W × H) of 50 mm × 25 mm × 1000 µm, 50 mm × 30 mm × 1500 µm and 45 mm × 24 mm × 150 µm, respectively. The glass membrane was chosen to be slightly smaller than the PDMS elastomer replica to protect its fragile edges during handling of the device.

The dimension of the sensor microchannel is 28 mm in length, while the width, the length and the height of Pt100 ceramic substrates is 2.0 mm, 2.3 mm, and 0.65 mm, respectively. The micropump chamber is in the shape of a cylinder with the diameter of 8 mm. The fluidic connections are in the shape of hollow cylinders with the internal diameter of 3 mm, the external diameter of 7 mm and the height of 8 mm. The inner and outer diameter of the polymer tubing is 1.6 mm and 3.2 mm, respectively.

Crucial geometrical parameters such as the depth of the channel, the width of the channel and the distance between adjacent Pt100 elements were first investigated by numerical simulations in order to determine the influence of these parameters on sensitivity and measuring range of the device. This approach facilitated a final determination of structure geometry prior to thermal mass flowmeter prototype fabrication.

## 3. Simulation Model

The 3-D fully coupled electro-thermal-fluidic model was built in Comsol Multiphysics 5.2. Flowmeter behavior can be explained considering three different physical models coupled together: The device is locally heated up by a biasing electric current or by constant temperature boundary condition (Comsol module: electric current). Generated heat is dissipated via conduction and convection (Comsol module: heat transfer in solids). A convection efficiency changes according to a flow in the device (Comsol module: laminar flow).

In the following, a brief description of relevant equations, boundary conditions, material parameters and a solution strategy is given.

The fluid flow is described by a Navier–Stokes equation:
(1)$$\rho \frac{\partial v}{\partial t}+\rho (v\cdot \nabla )v=\nabla \left[-pI+\mu \left(\nabla v+{\left(\nabla v\right)}^{T}\right)\right]+F$$
where the left hand side represents contribution of a force acting on a differential volume of the fluid and an inertial force. Here, v is a fluid velocity, ρ density, p pressure, μ dynamic viscosity, *T* temperature, *I* identity matrix and *F* are the external forces applied to the fluid. The above equation describing conservation of momentum needs to be solved together with an equation of mass continuity, which for incompressible fluid reads:(2)∇·v=0

Generated thermal power is related to biasing current through the platinum resistor:(3)$$Q=\rho \cdot J^{2}$$
where Q is the power density per volume unit, ρ is the material resistivity as a function of T and J is the current density.

Heat distribution or variation in temperature profile is derived from generated power density Q and fluid velocity v via heat conduction partial differential equation:
(4)$$\varphi C_{p}\hat{u}\cdot \nabla T=\nabla \cdot (k\nabla T)+Q$$
where T denotes temperature, φ is the material density, $C_{p}$ is the heat capacity at constant pressure, $\hat{u}$ represent a versor normal to the surface and k is the thermal conductivity of domain materials.

Equations (1)–(4) are solved for fluid velocity, fluid pressure, temperature and electric potential.

The microchannel was modeled as a 1.5 mm wide and 28 mm long block with defined fluidic inlet and outlet. PDMS, supporting glass and covering glass were modeled as blocks inserted in larger block representing surrounding air (Figure 2 left). Outer walls of the air block were thermally insulated which can be represented by boundary condition:−n·(k∇T)=0, where n is a normal vector on the boundary and k is a thermal conductivity. Pt100 elements were modeled as ceramics blocks with dimensions of 2.3 mm × 2 mm × 0.65 mm (L × W × H) comprising on its upper side a 50 µm wide and 1 µm thick platinum meandered line (Figure 2 right). The meander length was adjusted to 39 mm for specified resistance of 100 ohms at 0 °C.

The temperature of inlet fluid was set to 293.15 K, while the outlet of the channel was defined as a “thermal outflow” in Comsol “Heat transfer in solids” module. Channel outlet flowrate in “Laminar flow” module was set to zero, while channel inlet flowrate was taken from preselected values in Comsol parametric sweep procedure (0 to 100 µL·min^−1^ with a step of 10 µL·min^−1^). Essential material parameters were taken from a Comsol library of [20] and are listed in Table 1 for the temperature of 300 K.

Three physics were coupled in numerical model in order to physically describe (i) a non-isothermal flow (laminar flow coupled with heat transfer), (ii) an electric heat source (electric currents coupled to heat transfer) and (iii) a temperature coupling (heat transfer coupled to electric currents). To enable setting of boundary conditions such as a current through the heater or a voltage applied on the heater, electrical terminals in Comsol “Electric current” module were additionally applied on the platinum meanders. Electrical terminals enabled computation of flowmeter output signals by using manually entered Wheatstone bridge equations under “Derived values—Global evaluations” setting window in Comsol. The main challenge was meshing of the microchannel and the platinum meander due to a high length and width vs. depth ratio. Therefore, a user controlled mesh was employed in order to manually increase mesh density in these areas. In order to compute a Wheatstone bridge output signal as a function of a predefined set of flowrates, a parametric sweep function with a fully-coupled direct stationary solver was employed.

## 4. Fabrication

The PDMS cast with embedded Pt100 elements was fabricated by a soft lithography employing a replica molding technique in combination with rapid prototyping embedding technique. The silicon mold for the PDMS cast was fabricated by a deep reactive ion etching (DRIE) using the Bosch technique. A rapid prototyping embedding technique was applied in the following way: The PDMS Sylgard^®^ 184 (Dow Corning Corporation, Midland, MI, USA) two-component kit consisting of a pre-polymer (base) and a cross-linker (curing agent) mixed at the ratio 10:1 was poured in the Si mold and degassed (Figure 3a).

In the phase before the PDMS polymerization occurred, the Pt100 elements were inserted and manually aligned along the mold microchannel structures under a stereomicroscope, with their flat housing sides directed downwards (Figure 3b). To facilitate an alignment process, dedicated markers along the microchannels were employed (see an inserted detail at the bottom of the Figure 3a).

During the polymerization of the PDMS cast the viscosity of the cast elastomer dropped significantly, which facilitated settling of the flat bottom surfaces of the inserted Pt100s to the flat surface of the mold. Due to a low elastomer viscosity in this thixotropic transition stage, the elastomer was squeezed out of the interface between silicon mold and sensor bottom surface area forming extremely thin membrane layers. In this low viscosity stage of PDMS polymerization, dynamic fluctuations of the uncured elastomer were observed which were able to move prealigned Pt100 elements. The dynamic fluctuations of the PDMS elastomer during the transition stage were minimized by reducing the initial curing temperature from standard 60 °C to an ambient temperature for 24 h. Final curing was made at 80 °C for 2 h.

After curing, the PDMS replica with embedded Pt100 elements was cut into individual chips and punched by a biopsy puncher to realize through holes for fluidic connectors. Oxygen plasma activation was employed to irreversibly seal the PDMS cast with the glass substrate and the thin glass membrane. The fluidic connections were realized by means of covalently bonding the pre-punched PDMS hollow cylinders onto the supporting glass and by inserting the Tygon^®^ ND 100-65 tubing into the hollow cores of the cylinders. The micropump was driven by the disc-shaped PZT actuator (S-51, Sunnytec Electronic Co., Ltd., No. 9 Luke 1st. Road Lujhu Township, Kaohsiung County, Taiwan) that was glued on the top of the glass membrane by a conductive silver paste (EPO-TEK^®^ EE129-4, Epoxy Technology Inc., 14 Fortune Drive Billerica, MA, USA). Pt100 electrical connection wires were soldered to a header strip connection board that was glued firmly on the thick supporting glass enabling safe and robust handling. Figure 4a shows a schematic illustration of microchannel vertical cross-section together with the Pt100 heating/sensing elements. Please note that the surfaces of the Pt100 elements are wall mounted to the microchannel (marked in blue).

Figure 4b shows a photograph of the fabricated microfluidic chip comprising the thermal mass flowmeter and the integrated micropump with the electric and the fluidic connections.

## 5. Characterization

The fabricated flowmeter was characterized with a DI water flow by employing an automated measuring system [22]. Shortly, a custom-built high-voltage square waveform generator connected to a PC and a dedicated computer software were employed to automatically drive the micropump and simultaneously save instantaneous reference flowrate data and flowmeter output signal into a computer file. An excitation signal amplitude at a constant frequency of 50 Hz was swept automatically from 30 to 250 V at a step of 20 V every 60 s. An instantaneous water flowrate was defined using a gravimetric method (*Q* = (*dm*/*dt*) *ρ*) by employing a precision scale Kern ABJ 120-4M connected to a PC. A flowmeter output signal was led to a PC via a digital multimeter (Keithley 2700).

## 6. Results

We have designed a rapid prototyped thermal mass flowmeter based on commercial RTD Pt100 sensors and a custom made micropump [22]. In the following Section 6.1, numerical simulations were conducted in order to determine the influence of the crucial geometrical and thermal parameters on sensitivity and measuring range of the device. Based on the simulation results, crucial parameter values were set and a prototype device was fabricated and characterized. The characterization results are given in Section 6.2.

### 6.1. Investigation of Crucial Flowmeter Design Parameters by Numerical Simulation

For this set of simulations, a calorimetric constant temperature sensor-heater-sensor (S-H-S) flowmeter configuration was employed. This configuration was chosen as the most promising one, but it is also one of the most demanding approaches for practical implementation [23]. In the S-H-S configuration, the middle Pt100 element was employed as a heater while the first (upstream) and the third (downstream) were employed as temperature sensor elements and were connected in a Wheatstone bridge (Figure 5). Upstream and downstream sensors are defined relating to the flow direction.

Three geometrical parameters and a heater temperature parameter were varied one at the time through a specified range by using a parametric sweep procedure defined by Comsol.

The default parameter values for the channel width, the distance between adjacent platinum sensors, the channel depth and the heater temperature were set to 2 mm, 2 mm, 20 µm and 50 °C, respectively. For each value of the varied crucial parameters, a Wheatstone bridge output voltage was calculated for the flowrates ranging from 0 to 100 µL·min^−1^ with a step of 10 µL·min^−1^. The Wheatstone bridge supply voltage was set to 1 V.

Firstly, a channel width parameter was varied from 0.5 mm to 3 mm, keeping the distance between adjacent Pt100 elements, the channel depth and the heater temperature at the default values as defined in the above paragraph (Figure 6).

We define the upper limit of the measuring range as the flowrate value at which the flowmeter reaches 95% of its maximal response capability. Taking this in account, the upper limit of the measuring range for 0.5 mm wide measuring channel is 40 µL·min^−1^. At flowrates larger than 40 µL·min^−1^, the output voltage becomes saturated to a value of 7.5 mV. By increasing the channel width, the flowmeter upper limit of the measuring range gradually increases and reaches 100 µL·min^−1^ for 3.0 mm wide measuring channel. However, the widening of the channel affects the flowmeter sensitivity. To make this device comparable with similar devices already reported in the literature, sensitivity will be expressed in °C·µL^−1^·min units in order to be independent of the RTD nominal values and of the readout electronics. In the lower flowrate range from 0 to 10 µL·min^−1^, the sensitivity of the sensor decreases from 1.3 °C·µL^−1^·min to 0.625 °C·µL^−1^·min for 0.5 mm and 3 mm wide channel, respectively. By widening the channel, we speculate that two mechanisms counteract each other which affects the final flowmeter response. On one hand, by widening of the channel, the thermal transfer from the heater to the fluid and from the fluid to the sensors is promoted due to an increased contact surface area. But on the other hand, the total heat transfer is limited due to a lower fluid velocity in a wider channel. It is assumed that the second mechanism prevails, resulting in the decrease of the flowmeter response when the channel width is increased.

For the rapid prototyped mass flowmeter fabrication, the dedicated markers are crucial for proper manual alignment of Pt100 elements along the microchannel (see Figure 3a). The final channel width was set to 1.5 mm which enabled us to design the markers of sufficient size for precise manual alignment of Pt100 elements (see Section 2, Design). According to numerical simulation results (Figure 6), chosen channel width of 1.5 mm should assure flowmeter sensitivity of 1 °C·µL^−1^·min and the upper limit of the measuring range of more than 70 µL·min^−1^.

Secondly, the distance between adjacent Pt100 elements was varied from 0.1 mm to 2.5 mm, whereby the channel width was set to 1.5 mm and the channel depth and the heater temperature at default value. Simulated results are shown in Figure 7. By increasing the distance between adjacent Pt100 elements, the flowmeter upper limit of the measuring range gradually decreases according to the criteria set in Section 6.1. At the same time, the sensitivity in the lower flowrate range (from 0 to 10 µL·min^−1^) increases, from 0.55 °C·µL^−1^·min to 1.2 °C·µL^−1^·min for spacing distance of 0.1 mm and 2.5 mm, respectively. We observed that increasing of the distance between adjacent Pt100 elements has an analogous effect on the Wheatstone bridge output response as narrowing of the fluidic channel.

By increasing the distance between adjacent Pt100 elements at non-zero flowrate, a temperature difference between upstream and downstream sensing element is increased due to a non-symmetric thermal distribution profile of a flow stream, which is shifted in the direction of the flow, increasing flowmeter sensitivity. However, an increased distance between adjacent elements might affect flowmeter time response at low fluid velocities, since more time is needed to transfer the heat from the heater to the distant placed sensor elements at sudden flowrate changes. Furthermore, by increasing the distance between adjacent elements, the upper limit of the flowmeter measuring range decreases. In this respect, we set the distance between adjacent Pt100 elements to 1.5 mm (a compromise value) for the fabricated prototype.

The symmetrical configuration is not the most efficient configuration in order to obtain the optimal measuring range, since the contribution of each sensing element (upstream and downstream) to the total sensor signal is not the same. For further optimization, the distance of each element should be studied separately and the optimal up/downstream location should be extracted in order to obtain the widest flow range. An excellent work in terms of this approach was undertaken by Petropoulos A. and Kaltsas G. in 2010 [1]. The authors managed to increase the measuring range of their PCB-MEMS liquid microflow sensor by setting the upstream distance to an optimum value of 0.425 mm and by increasing the downstream distance at the same time. They concluded that no specific combination of the sensing elements distance provides optimal results throughout the entire measuring range, which implies that the flowmeter design should be an application specific.

Thirdly, the channel depth parameter was varied from 10 µm to 1000 µm keeping the channel width and the distance between adjacent Pt100 elements at selected values of 1.5 mm and 1.5 mm, respectively. Heater temperature was set to the default value. Simulated results are shown in Figure 8.

After increasing the channel depth from 10 µm to 20 µm, no significant alteration in the sensor output characteristic was observed. Both characteristics increase exponentially to a maximum of 7.8 mV. The steep slope in an initial flowrate interval of 0 to 10 µL·min^−1^ can also be described by a linear function with a slope coefficient of 52 × 10^−5^ V min·µL^−1^ (equals 1.3 °C·µL^−1^·min). At the channel depth of 100 µm, the bridge output voltage increases exponentially to a maximum value of 7.46 mV at 60 µL·min^−1^ and then gradually decreases to 7.28 mV at 100 µL·min^−1^. As the channel depth increases further, the bridge response decreases, which affects both the maximum output voltage and the slope coefficients, i.e., the sensitivity. With a channel depth parameter value of 1000 µm, the sensitivity of the flowmeter decreases to 0.7 °C·µL^−1^·min (46% drop) and the maximum response to 4.342 mV (44% drop) as compared to 10 µm channel depth.

An increase of the channel depth parameter value did not extend the initial measuring range of the device as one might expect due to the presumption that deeper measuring channel allows more fluid flow through the sensor at the same heat transfer from the heater to adjacent sensor elements.

Indeed, when the focus is on the limited flowrate interval from 0 to 20 µL·min^−1^ (Figure 8), the output signal for the same flowrate does decrease with the increasing depth of the channel. From this correlation, one might wrongly assume that the deeper channel sensor characteristic will saturate latter and extend the upper limit of the sensor measuring range. On the contrary, simulation results reveal that the initial upper limit of the measuring range is rather decreased due to inflection points in Wheatstone bridge output characteristics (for channel depth ≥100 µm, as seen in Figure 8).

The parameter that is actually measured by the device is the flow velocity over the heater/sensor elements and not the volumetric flowrate. However, even when the results are presented as a function of the mean flow velocity for various channel depths (Appendix A), an implementation of a deeper channel does not extend the measuring range of the mean flow velocity.

In order to study this phenomenon in detail, additional numerical simulations were performed. For two extreme cases of the channel depth i.e., 10 µm and 1000 µm, the temperature vs. flowrate relation for the upstream and the downstream sensor element was studied in simulation environment where the flowrate was varied from 0 to 100 µL·min^−1^ for both channel depths (Figure 9).

In both channels (depth of 10 and 1000 µm), less and less heat is transferred to the upstream element as the flowrate increases, which reflects in the temperature drop of the upstream element (Figure 9, squares and circles symbols).

It is even more significant to study the temperature of the downstream sensing element. In order to explain the inflection points of sensor characteristics (see Figure 8) a maximum temperature difference of the medium beneath the heating element and beneath the downstream element was studied in the simulation environment (a vertical temperature profile of the medium beneath the elements). The results of the study follow below:

With a shallow 10 μm channel, the heater heats the medium down to the bottom of the channel regardless of the flowrate (10 to 100 µL·min^−1^), and the maximum temperature difference of the medium under the heater is only 0.0116 °C and 0.0175 °C for the flowrate of 10 µL·min^−1^ and 100 µL·min^−1^, respectively. This is due to the high ratio between the surface of the heater and the depth of the channel.

The temperature under the downstream sensor is also vertically uniform through the depth (<0.002 °C difference for all flowrates) and increases with the flowrate, which means that the higher the flowrate, the more heat is transferred from the heater to the downstream element. This is indicated by the elevated temperature of the downstream element (Figure 9, triangle symbols). Since the output signal is proportional to the temperature difference of the two sensing elements, the signal follows the increase in flowrate in the range of 0 to 100 µL·min^−1^.

For the case of the 1000 µm deep channel, at a flow rate of 10 μL·min^−1^ (see Figure 10a), the heater is sufficiently efficient despite the poor surface vs. depth ratio. This is due to the low medium velocity and the maximum temperature difference of the medium under the heater still does not exceed 2.97 °C. As a result, even under the downstream sensor element, the medium temperature is quite uniform through the depth (<0.11 °C difference) and accounts for 48.44 °C in proximity of the element.

However, when the flowrate parameter is increased (see Figure 10b), the heater can no longer heat the medium near the bottom of the channel. Therefore, at a flow rate of 100 µL·min^−1^, the medium temperature at the bottom of the microchannel is as much as 23 °C lower (*T* = 27 °C) as compared to the position right beneath the heater.

The lower cooler layer of the medium cools the warmer upper layer of the medium on its way toward the downstream element due to the vertical heat conduction. As a result, the temperature of the upper layer of the medium on its way to the downstream element already drops sharply (from 48.8 °C beneath the heater to 39 °C beneath the sensor for 100 µL·min^−1^). This is reflected in the graph as the cooling of the downstream element by increasing the flowrate (Figure 9, diamond symbols). Cooling of the upper medium layer due to the cooler bottom medium layer for channels depth ≥100 µm and for flowrates ≥40 µL·min^−1^ is reflected in characteristic inflections (Figure 8) and thus limits the measuring range of the device. To overcome this deficiency, a flowmeter with two heater elements positioned on both sides of the channel might be used which would heat the flow more uniformly.

Similar flowmeter characteristics with inflection points were reported by Lammerink, T.S. et al. [24]. In the reported work, the authors derived the turn-over flow velocity $v_{to}$ (from a fluid convection analytical model) which determines the useful flow range of the device:
(5)$$v_{to}=\frac{2D}{l_{z}}$$
where $l_{z}$ is a measuring channel depth and *D* is a thermal diffusivity. According to their model, by defining a turn over flowrate as $\Phi_{to}=v_{to}\cdot l_{y}\cdot l_{z}=2Dl_{y}$, where $l_{y}$ is a channel width, the turn over flowrate $\Phi_{to}$ and thus an useful flow range of the device cannot be extended by the channel depth parameter $l_{z}$. However, according to their model, the measuring range of the device can be extended by widening of the channel ($l_{y}$), which again correlate with our simulation results (see Figure 6).

Based on simulation results, the depth of the channel for the fabricated prototype was set to 10 µm. This parameter value should extend the upper limit of the flowmeter measuring range and simultaneously provide suitable geometric conditions for realization of a single actuator peristaltic micropump [19] which is integrated on the same chip. However, with such a shallow depth of the microchannel, a relative non-uniformity of the depth along the microchannel is promoted. Non-uniformity of the depth can be caused by fluidic pressure which might deform the channel walls and the 150 µm thick glass membrane, or by stress on embedded Pt100 elements caused by e.g., electrical wiring. Furthermore, heating of the Pt100 elements causes temperature elongations of its ceramic base and surrounding PDMS which might cause further deformation of the channel.

Lastly, the heater temperature was varied from 30 °C to 80 °C keeping the crucial geometrical parameters i.e., the channel width, the distance between adjacent Pt100 elements and the channel depth at selected values of 1.5 mm, 1.5 mm and 10 µm, respectively. Simulated results are shown in Figure 11.

From simulated characteristics it is evident that increasing of the heater temperature increases flowmeter sensitivity while simultaneously maintaining the measurement range almost unchanged. Simulated sensitivity in the range of 0 to 20 µL·min^−1^ increased from 0.145 °C·µL^−1^·min to 1.75 °C·µL^−1^·min for the heater temperature of 30 °C and 80 °C, respectively. It should be mentioned that sensitivity of thermal mass flowmeter depends also on a medium temperature, since heat transfer from e.g., heater to the medium is driven by a temperature difference between both objects. However, investigation of this parameter is not covered in this work. To assure high sensitivity characteristics for our fabricated prototype in constant temperature S-H-S configuration we set the heater temperature to 80 °C (see Section 6.2.1). For volatile and low-temperature evaporation chemicals, the temperature of the heater should be set much lower. In this case, the loss of sensitivity could be compensated for by increasing the distance between adjacent Pt100 elements, but on account of a reduced measuring range.

### 6.2. Characterization of the Fabricated Prototype

The prototype was designed and fabricated on the basis of geometrical parameters given in Section 2 and crucial parameters, which were determined by employing numerical simulations in Section 6.1. Fabrication of the prototype is presented in Section 4. Crucial design and temperature parameters are summarized in Table 2.

Once the Pt100 elements are embedded in the PDMS elastomer along the channel, they can be connected in various modes of operation. In order to determine an optimal flowmeter configuration for our precise low flowrate application (i.e., flow controlled micropump) by means of an adequate sensitivity and required upper limit of the measuring range, the fabricated flowmeter was employed in three selected configurations which differ from each other in a measuring approach. The first is a calorimetric sensor-heater-sensor (S-H-S) configuration, the second is a hot wire H/S configuration and the third is a heater/sensor-heater/sensor (H/S-H/S) configuration.

#### 6.2.1. Sensor-Heater-Sensor (S-H-S) Configuration

The heater in S-H-S configuration can be employed in a constant voltage, a constant current, a constant power or a constant temperature regime. In this work, a constant voltage and a constant temperature regime was studied and analyzed. A wiring diagram of a constant voltage thermal mass flowmeter in S-H-S configuration is shown in Figure 12.

A sensor elements self-heating was minimized by keeping a measuring current below 1 mA using two 1 kΩ resistors and a bridge input voltage of 1 V. Self heating was measured separately on one sensor element with primed channel at a room temperature by using Keithley 2700 multimeter. The temperature of the heater was measured before and immediately after five minutes of applied bridge voltage of 1 V. The temperature rise due to the self heating was below the limit of measurability for our setup.

During measurements, a voltage of 5 V was applied to the heating element to ensure a heating temperature in the range of 60 to 80 °C.

The Wheatstone bridge output signal as a function of the flowrate for constant voltage S-H-S configuration is shown in Figure 13. The device shows response, which could be well fitted by an exponential rise to maximum mathematical function. At flowrates below 20 µL·min^−1^, the device yields almost linear response with the sensitivity of 0.518 °C·µL^−1^ min and reaches saturation at 80 µL·min^−1^. At flowrates higher than 80 µL·min^−1^ the response decreases with the increasing flowrate. This is due to heater temperature drop owing to convective cooling at higher flow rates. This means that the temperature gradient upstream and downstream decreases, which in turn decreases sensor output signal.

Due to measured signal saturation, the upper limit of the measuring range in this configuration is estimated to be 40 µL·min^−1^. Simulated results yield slightly higher sensitivity than the measured results (0.625 °C·µL^−1^·min) and saturates to 2.36 mV at 60 µL·min^−1^ which is in good agreement with measured results.

In order to extend the upper limit of the flowmeter measurement range, the measuring procedure was repeated by ensuring a constant heater temperature. This regime of operation was simulated in detail in Section 6.1.

To provide the constant heater temperature regime, the applied voltage was tuned manually for each flowrate to yield a constant *U_heater_*/*I_heater_* ratio, which corresponds to a constant heater resistance and thus the constant heater temperature. For practical use, dedicated electronics for automatic heater temperature regulation would be more appropriate. The advantage of this approach is shown in Figure 14. An extended flowrate measuring range from 60 to 90 µL·min^−1^ in respect to the constant temperature of the heater was obtained.

Flowmeter output signal as a function of flowrate at the constant temperature of 80 °C (*R_heater_* = 130 Ω) is shown in Figure 14. In the simulation environment, the constant heater temperature was assured by setting all heater boundaries’ temperature to 80 °C in a Comsol “heater transfer in solids” module.

In this approach, the flowmeter yielded an exponential growth response for flowrates up to 110 µL·min^−1^. Using the same criterion as in Figure 13, the upper limit of the measurement range was estimated to be 90 µL·min^−1^. Steep slope in an initial flowrate interval of 0 to 40 µL min^−1^ may also be described by a linear function with a slope coefficient of 51.9 × 10^−5^ V·µL^−1^·min (equals 1.3 °C·µL^−1^·min), which corresponds to flowmeter sensitivity parameter. Simulation results confirm extension of the measuring range by maintaining the constant heater temperature. The discrepancies between measurements and simulation can be ascribed to a consequence of several factors starting with a non-ideal fabrication process, physical and geometrical model simplifications, selected simulation material parameters, selected mesh density or solver tolerance.

#### 6.2.2. Heater/Sensor (H/S) Configuration

In a hot wire H/S configuration, only one Pt100 element (the second downstream sensor) was employed, performing two functions; i.e., heating and sensing. The element was biased at a constant voltage of 5 V. Medium flowrate was determined by measuring the current through the element (Figure 15).

In this configuration, the measuring range of the flowmeter is greatly extended as compared to previous two configurations (see Figure 16). Measured current characteristic curve exhibit quadratic growth from initial 37.3 mA at zero flowrate to 41 mA at maximal achievable flowrate of 112 µL·min^−1^. This corresponds to the heater temperature of 88 °C and 57 °C, respectively. By increasing the flowrate, the heat transfer from the heater is increased. The temperature of the heater drops, which in turn reduces heater resistance. Due to the constant voltage supply and the reduced heater resistance, a heating electrical power increases and thus partly compensate a heater cooling. The flowmeter yielded sensitivity of 0.02 mA µL^−1^ min in the flowrate range of 0 to 40 µL·min^−1^.

Simulated results show similar characteristic trend, but at slightly lower absolute values (<2.5% @ maximal flowrate). The discrepancies between measurements and simulation are the results of many factors as described in Section 6.2.1.

The shortcomings of this configuration include requirement for high precision ampere meter and inability to determine flow direction.

#### 6.2.3. Heater/Sensor-Heater/Sensor (H/S-H/S) Configuration

In the H/S-H/S configuration, the first and the third Pt100 elements in downstream were employed, where each was acting as a heater and as a sensor (Figure 17). In this configuration, both elements are connected in a Wheatstone bridge. The constant current was set to 35 mA to heat both elements to 80 °C at zero flowrate and the Wheatstone bridge output signal was measured as a function of the flowrate as shown in Figure 18.

Measured and simulated results clearly show that such a configuration provided high linearity with sensitivity of 0.613 °C·µL^−1^·min in the measuring range of 0 to 60 µL·min^−1^. By increasing the flowrate above 60 µL·min^−1^ the temperature of the elements decreased sharply which prevented sufficient heat transfer from the first element to the downstream one. The simulation results are in a correlation with the measured results, yielding the similar slope coefficient in the range of 0 to 60 µL·min^−1^ and similar descending trend at highest achievable flowrate.

Comparing this configuration with the previous two, the following conclusions can be drawn:

In the S-H-S configuration and the H/S-H/S configuration, the upper limit of the measuring range did not exceed 60 µL·min^−1^ due to the enhanced flow induced cooling of the heating elements which consequently prevented sufficient heat transfer between adjacent elements. However, when a heater temperature was maintained constant, the measuring range in S-H-S configuration extended up to 90 µL·min^−1^. Simplest measuring configuration H/S employed only one element. For the flowmeter in such a configuration, the measuring range was significantly extended but as such it yielded low sensitivity of 0.02 mA·µL^−1^·min in the range of 0 to 40 µL·min^−1^, required precision measuring equipment, and could not demonstrate distinct flow direction. Overall, a constant temperature S-H-S flowmeter configuration resulted in an adequate measuring range of 0 to 90 µL·min^−1^ and high flowmeter sensitivity of 1.3 °C·µL^−1^·min in the lower flowrate range of 0 to 40 µL·min^−1^ and so is the most appropriate for precise low flowrate applications. To measure higher flow rates, the mass flowmeter should be employed in a bypass structure with the main flow.

## 7. Conclusions

An innovative rapid prototyping technique for embedding micro components into PDMS replicas is presented. Micro components are inserted in casted PDMS and aligned on a mold. After PDMS polymerization, they become embedded in the cast, ready to perform additional functions in the chip. The technique was applied on a thermal mass flowmeter that is suitable for implementation into microfluidic systems that require flow control of reagents, but need to be fabricated in a rapid and a cost efficient way and do not necessarily require high accuracy and a fast response time of conventional MEMS micro flowmeters. To further facilitate fabrication process, commercially available miniature RTD Pt100 components were employed as sensing and heating elements.

Influence of crucial design and thermal parameters on a sensitivity and a measuring range of the device were investigated by a 3D fully coupled electro-thermal-fluidic model in Comsol Multiphysics 5.2 simulation software.

Contrary to our expectations, the simulation results show that implementation of a deeper channel does not increase the upper limit of the measuring range. Additional simulations revealed that a non-uniform temperature profile beneath the heater plays a major role, causing the cooling of the upper medium layer that is involved in a heat transfer from the heater to the sensing elements.

Based on the simulation results, a thermal mass flowmeter prototype was fabricated together with a micropump to monitor and/or control the micropump flowrate and it was characterized for three distinct measuring configurations.

For precise low flowrate applications, a S-H-S flowmeter configuration with a constant heater temperature was found as the most appropriate one assuring a measuring range of 0 to 90 µL·min^−1^ and sensitivity of 1.3 °C·µL^−1^·min in the lower flowrate range of 0 to 40 µL·min^−1^.

## Figures and Tables

**Figure 1 sensors-21-05373-f001:**
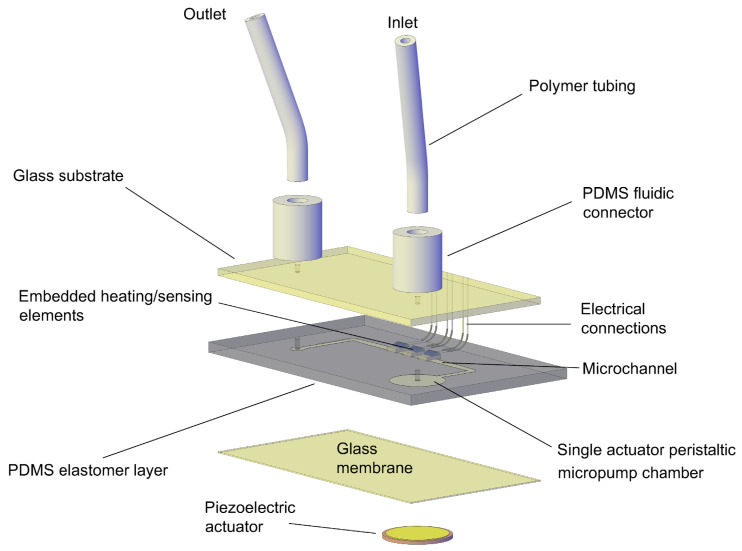
An exploded view drawing of the microfluidic chip comprising the piezoelectric micropump and the micro flowmeter in the microchannel.

**Figure 2 sensors-21-05373-f002:**
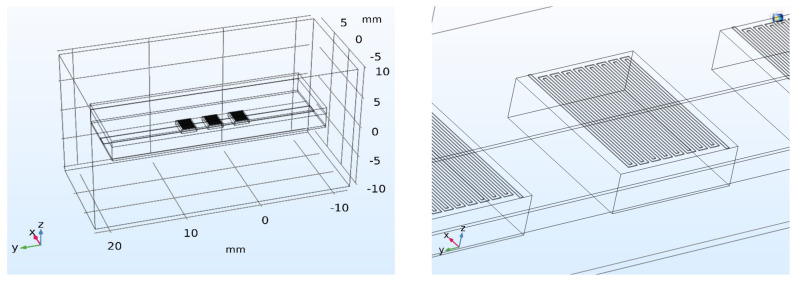
A geometry of the analyzed structure in the simulation environment and (**left**) a detail of simulated RTD Pt100 elements with integrated platinum meander structure (**right**).

**Figure 3 sensors-21-05373-f003:**
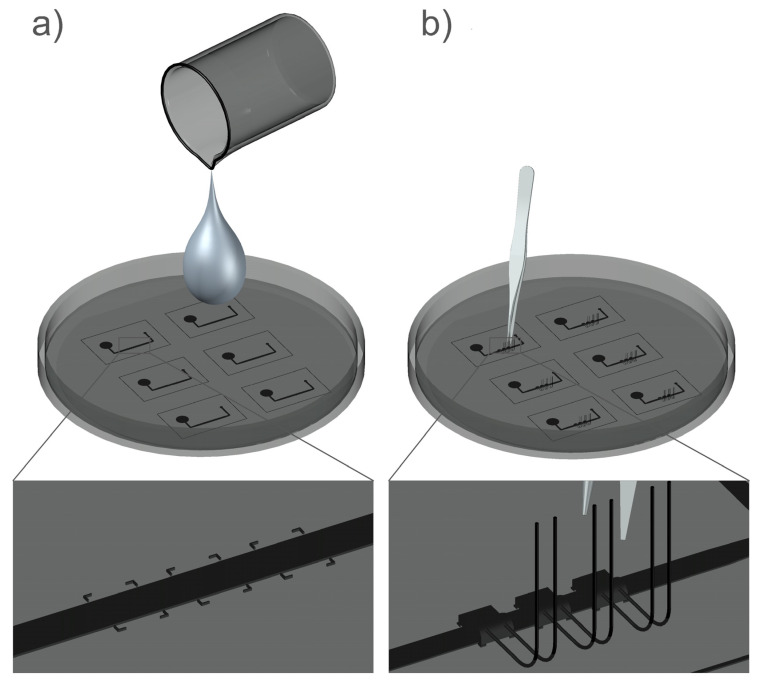
Two crucial steps of the rapid prototyping embedding technique for a flowmeter fabrication: (**a**) pouring of the degassed PDMS elastomer onto the Si mold; (**b**) manual alignment of platinum elements to the dedicated markers along the microchannel.

**Figure 4 sensors-21-05373-f004:**
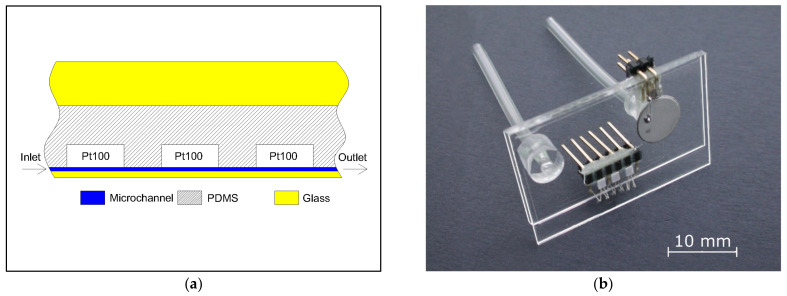
(**a**) A microchannel vertical cross-section of the thermal mass flowmeter with wall mounted Pt100 heating/sensing elements. Dimensions are not to scale; (**b**) a photograph of the fabricated microfluidic chip comprising the thermal mass flowmeter and the micropump.

**Figure 5 sensors-21-05373-f005:**
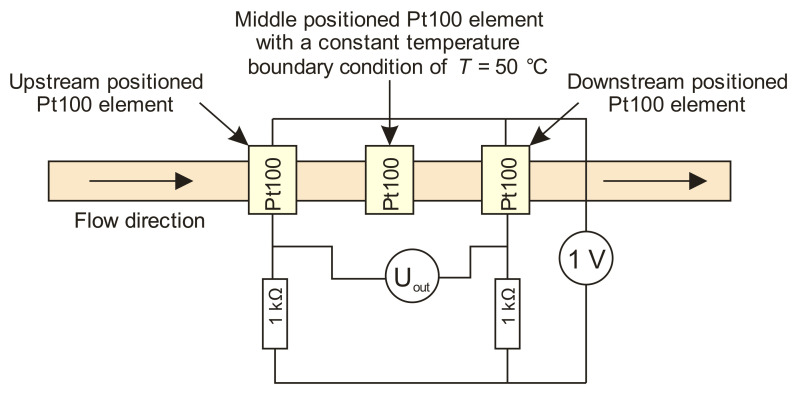
A wiring diagram of a constant temperature thermal mass flowmeter in S-H-S configuration which was implemented in a Comsol simulation model.

**Figure 6 sensors-21-05373-f006:**
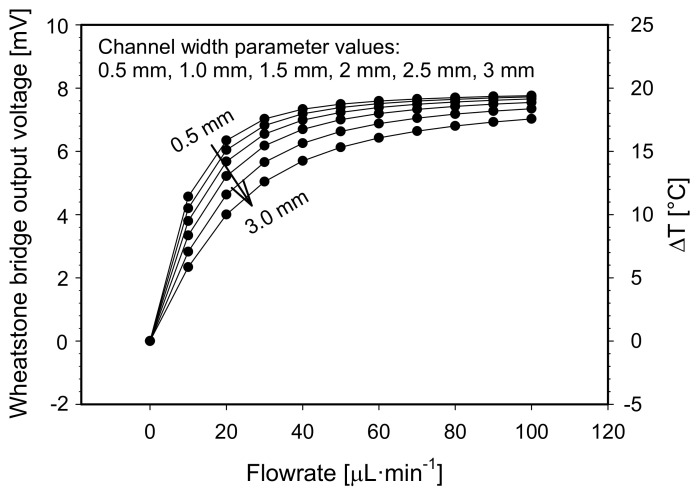
Constant temperature S-H-S flowmeter configuration: simulated Wheatstone bridge output voltage and differential temperature between both sensor elements vs. flowrates for preselected channel width parameter values.

**Figure 7 sensors-21-05373-f007:**
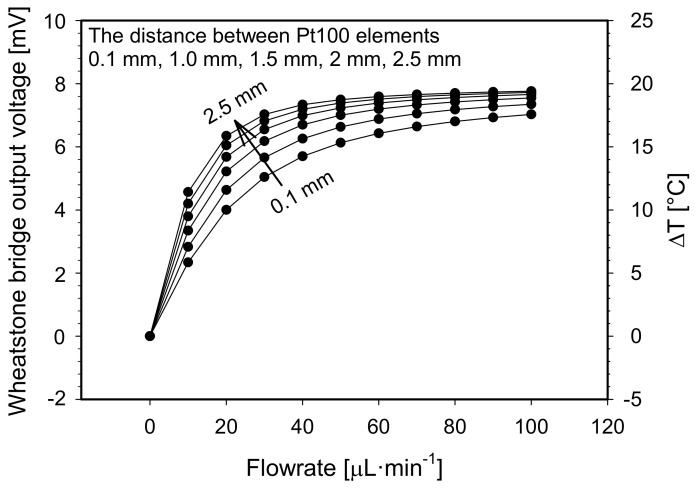
Constant temperature S-H-S flowmeter configuration: Simulated Wheatstone bridge output voltage and differential temperature between both sensor elements vs. flowrate for preselected distances between adjacent Pt100 elements.

**Figure 8 sensors-21-05373-f008:**
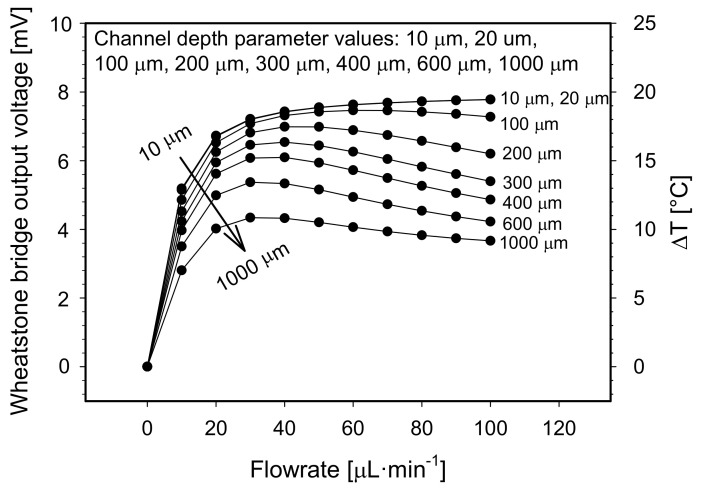
Constant temperature S-H-S flowmeter configuration: simulated Wheatstone bridge output voltage and differential temperature between both sensor elements vs. flowrates for preselected channel depth parameter values.

**Figure 9 sensors-21-05373-f009:**
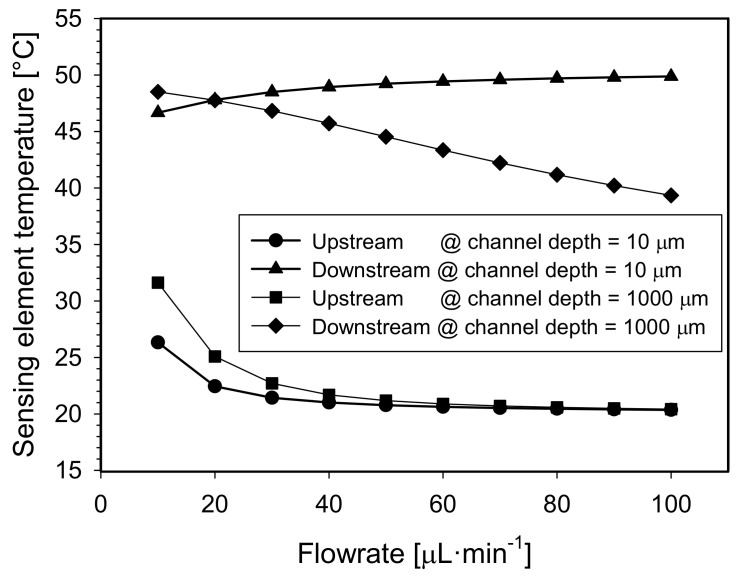
A simulated temperature of the upstream and the downstream sensor element vs. flowrate for two extreme cases of the channel depth (10 µm and 1000 µm).

**Figure 10 sensors-21-05373-f010:**
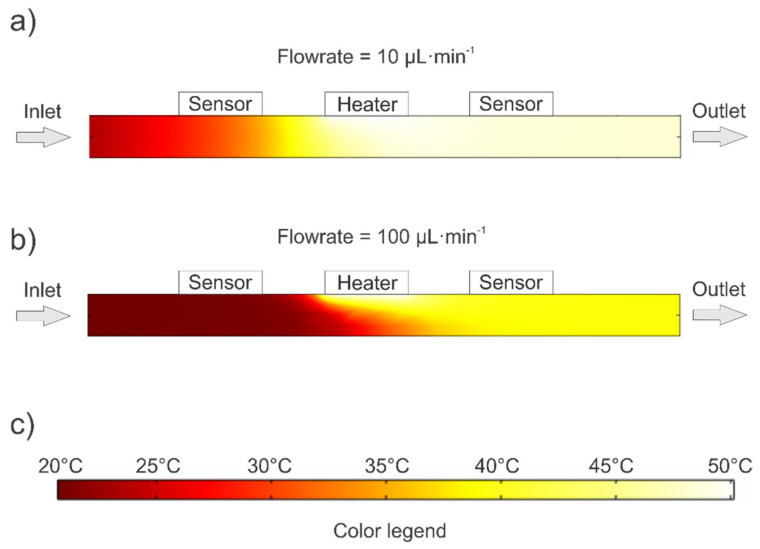
A simulated temperature distribution in a cross section along the 1000 µm deep channel (**a**) for flowrate of 10 µL·min^−1^ and (**b**) for flowrate of 100 µL·min^−1^. Color legend (**c**) is the same for both case studies.

**Figure 11 sensors-21-05373-f011:**
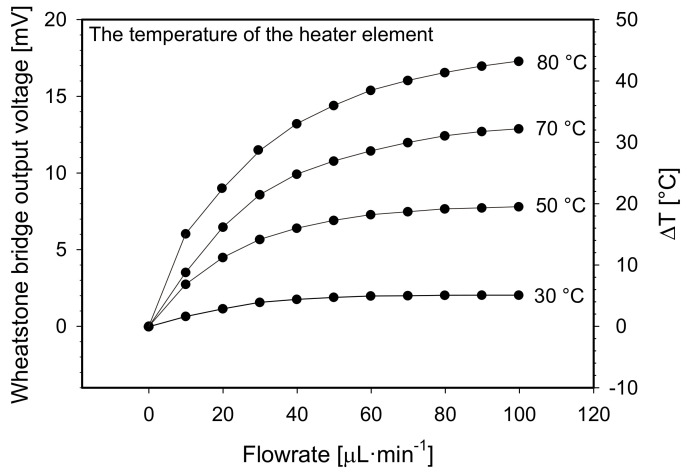
A constant temperature S-H-S flowmeter configuration: Simulated Wheatstone bridge output voltage and differential temperature between both sensor elements vs. flowrate for preselected heater temperature values.

**Figure 12 sensors-21-05373-f012:**
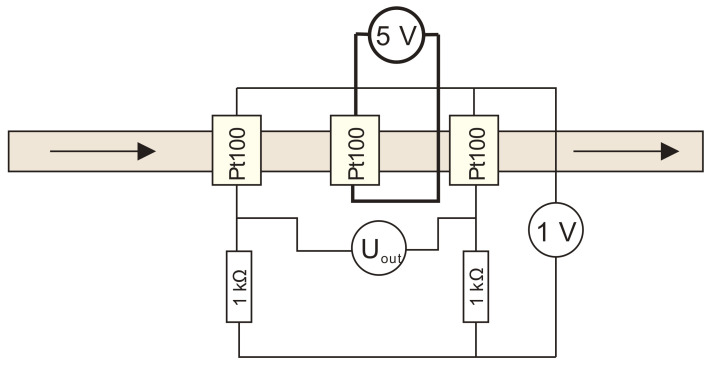
A wiring diagram of a constant voltage thermal mass flowmeter in S-H-S configuration.

**Figure 13 sensors-21-05373-f013:**
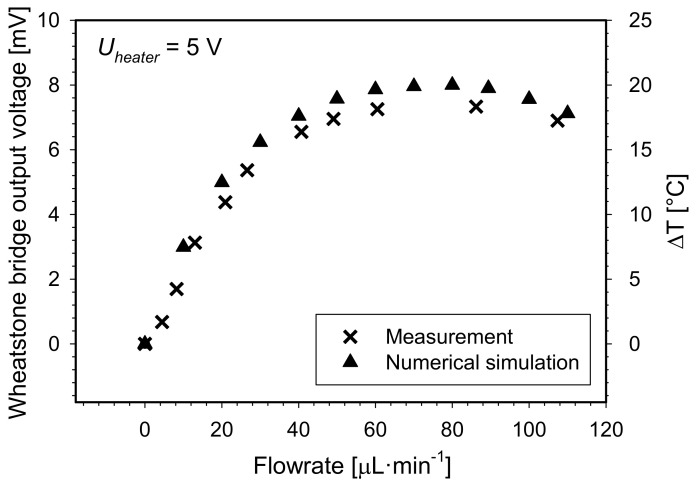
Measured and simulated Wheatstone bridge output signal and differential temperature between both sensor elements vs. flowrate for S-H-S configuration with a constant heater supply voltage.

**Figure 14 sensors-21-05373-f014:**
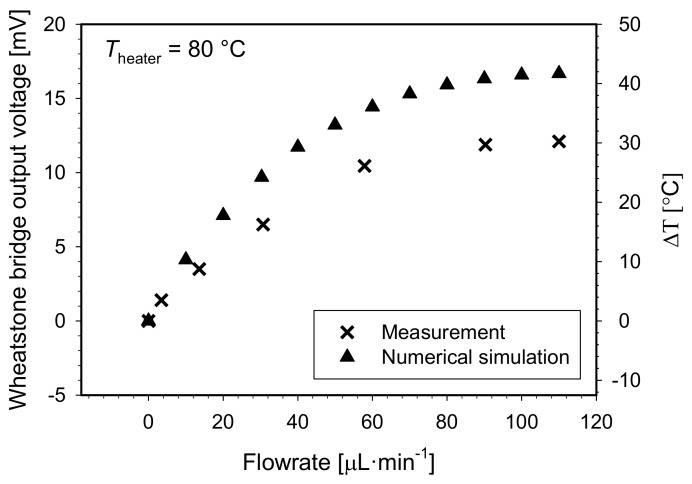
Measured and simulated Wheatstone bridge output signal and differential temperature between both sensor elements vs. flowrate for S-H-S configuration under constant heater temperature regime.

**Figure 15 sensors-21-05373-f015:**
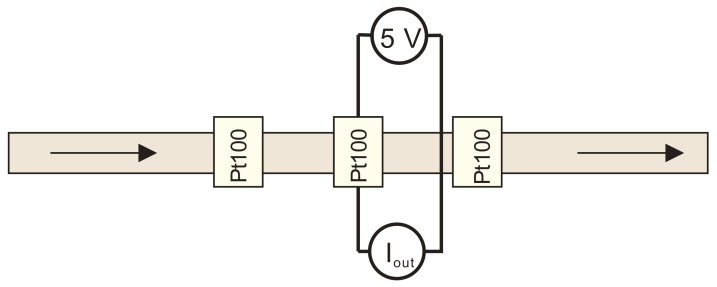
A wiring diagram of the thermal mass flowmeter in a heater/sensor configuration.

**Figure 16 sensors-21-05373-f016:**
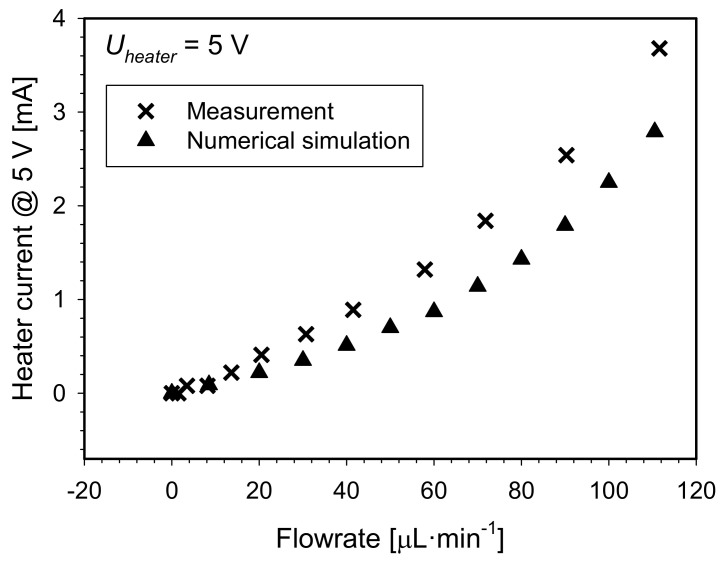
Measured and simulated heater current versus flowrate for H/S configuration.

**Figure 17 sensors-21-05373-f017:**
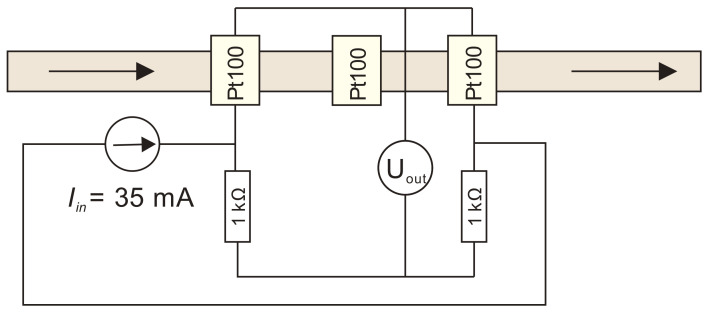
A wiring diagram of a thermal mass flowmeter in the heater/sensor-heater/sensor configuration.

**Figure 18 sensors-21-05373-f018:**
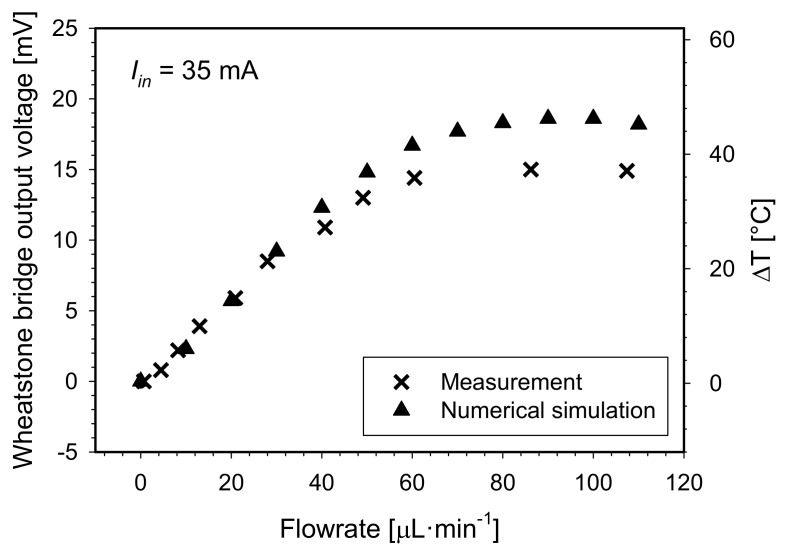
Measured and simulated Wheatstone bridge output signal and differential temperature between both sensor elements vs. flowrate for the H/S-H/S configuration at constant current heating.

**Table 1 sensors-21-05373-t001:** List of material properties at *T* = 300 K that were required by three physical models in the Comsol simulation model.

	Thin Film Platinum	Ceramics	Water	Glass	PDMS ^2^	Air
Density [kg·m^−3^]	21,450	2600	1000	2210	970	1.16 ^1^
Heat capacity at constant pressure [J·kg^−1^·K^−1^]	133	850	4200	730	1460	1010
Thermal conductivity [W·m^−1^·K^−1^]	71.6	2.9	0.61	1.14	0.16	0.025
Electrical conductivity [S·m^−1^]	9.43 × 10^6^ [21]					
Temperature coefficient [K^−1^]	0.0045 [21]					
Dynamic viscosity [Pa·s]			8 × 10^−4^			

^1^ at 10^5^ Pa. ^2^ Polydimethylsiloxane elastomer.

**Table 2 sensors-21-05373-t002:** Crucial design and temperature prototype parameters.

Channel Width	1.5 mm
Distance between adjacent Pt100 elements	1.5 mm
Channel depth	10 µm
Temperature of the heater	80 °C

## Data Availability

Not applicable.

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
