# Peer review of "A Rapid Prototyped Thermal Mass Flowmeter"

_sensors, 2021, doi:10.3390/s21165373_

Round 1
Reviewer 1 Report
This work reports on the fabrication and evaluation of a thermal flow meter, implemented by embedment of RTDs in PDMS. A detailed FEA study is also reported for the operation of the proposed device in various modes.
Although the main aspects of the work are not innovative since several similar devices have already been reported in the literature by various groups, my view is that this work is solid, well presented and justified. Thus, I believe that it can be published with the below mentioned remarks:
- The sensitivity of the device, in all the cases (except form H/S case) is expressed in V/(l/min), which according to the presented evaluation, represents the output of the Wheatstone bridge as a function of the flow. The corresponding response is determined by the readout electronics and not by the device itself, thus it cannot be comparable with similar devices already reported in the literature. I believe that the authors should use a more global representation of the sensitivity like: ΔR/(l/min), which indicates the device performances or even better ΔΤ/(l/min), which is not depended on the RTD nominal value. Additionally, I strongly suggest a normalized sensitivity expression (ΔΤ/(l/min))/Input_Power which is independent from the applied power to the heating element as well, and is directly comparable in all the cases.
- Lines 64-66: “Although a PDMS is frequently employed in flow-meter applications, to the best of our knowledge, this is the first time that a PDMS is employed as a heat insulative material between heater/sensor elements”
This is not the case, since many works utilized PDMS for flow sensor fabrication (e.g.: Murilo Z. Mielli et. al. “Thermal flow sensor integrated to PDMS-based microfluidic systems”, DOI: 10.1109/SBMicro.2013.6676148
Haixia Yu et. al. “A micro PDMS flow sensor based on time-of-flight measurement for conductive liquid”, DOI 10.1007/s00542-012-1686-7)
- The authors mention that “The dimension of the sensor microchannel is 28 mm in length”, while in section “3. Simulation Model” they state that “The microchannel was modeled as a 1.5 mm wide and 19 mm long block”. In figure 2 it seems that the simulated microchannel length is close to 30mm. The authors should explain this mismatch.
- According to the theoretical study the authors choose a microchannel depth of 10μm. It is not clear from the fabrication process if the surface of the Pt100 element is wall mounted to this microchannel or the microchannel depth is 10μm + Pt100 thickness. In the second case the disturbance of the flow due to the Pt100 profile should be discussed. In any case the microchannel depth is very low comparing to the Pt100 thickness (650μm), thus more details regarding the implementation of these devices into the microchannel needed. The authors should present a photo of the cross-sectional area of the microchannel above the Pt100 element. Moreover, given the fact that the length of the microchannel is about 30mm and PDMS is an elastomer material, the uniformity of the depth along the microchannel should be discussed, with more focus on the RTD locations.
- During the theoretical evaluation regarding the optimum distance between adjacent Pt100 elements, the authors assume symmetrical sensing element location. This is not the most efficient configuration in order to obtain the optimum flow range, since the contribution of each sensing element (upstream and downstream) to the total sensor signal is not the same; therefore, the distance of each element should be studied separately and the optimum up/downstream location should be extracted in order to obtain the widest flow range. Furthermore, the location of the sensing elements is a function of the heater temperature, since that is the dominant parameter for the formation of the zero-flow temperature field; thus, these two parameters should be studied together and not independently. For example, if a different heater power chosen, then the optimum up/downstream sensing element distances will change respectively.
- Regarding the evaluation of the microchannel depth in figure 8, the sensor response is illustrated as a function of flowrate for various channel depths. It should be underlined that the parameter that is actually measured by this device is the flow velocity (over the RTDs) and not the flowrate. By increasing the channel depth, the flow velocity decreases (with the same flowrate), thus the response of the sensor drops. The authors should present the response of the sensor as a function of mean flow velocity with various channel depths in order to compare the corresponding responses. Moreover, an extra figure should be added with the temperature distribution in a cross section along the channel, in order to support the explanation of temperature difference between lower and upper layer.
- In section 6.2.2 it is noted that “The shortcomings of this configuration include an absence of temperature compensation”. The same notation is repeated in line 538. This is not true since temperature compensation is common in hot-wire mode. In this case the heater is kept at a constant temperature and the sensor signal is defined by the applied power, which is required to maintain the heater at the same temperature, for various flowrates. I suggest that the authors should implement this mode in their work and they’ll obtain interesting results.
Reviewer 2 Report
The authors have developed a mass flowmeter based on temperature sensors whose accuracy is suitable for lower ranges of flow. They have also performed ComSol simulations to develop a 3D model for their sensor. However, there are some questions and comments that need to be addressed before publication.
- What is the main novelty of this work? It looks like a system design and not a specific sensor design for a mass flowmeter. It may be a better fit for system-centric journals.
- Just as a comment, this kind of mass flowmeters can measure higher flow rates if used in a bypass structure with the main flow. In that case, the target application can be broadened to industrial applications.
- It is better to give a summary of the simulation optimization in the main text and move more detailed optimizations to the supplementary section. It is very hard to follow the story of this manuscript.
- Even though the 2 upstream and downstream sensors are just biased for sensing, there is no discussion about how the heat generated in these sensors will be canceled out in the post-analysis section.
- The authors mentioned that at higher flow rates and wider channels the heater may not be able to sufficiently heat the flow due to the lack of contact. As a suggestion, they can add another heater to the other sides of the channel to be able to heat the flow more uniformly.
- In Figure 13, there are some discrepancies between measurement and simulation. Can this discrepancy come from the constant high temperature and degrading in the device including the PDMS material?
- Using a high-temperature heater (like 80 C) may limit the application of this prototype as some chemicals will be evaporated at this temperature. Authors can explain how this prototype can be reconfigured for those volatile and low-temperature evaporation chemicals.
Round 2
Reviewer 1 Report
The authors addressed all my remarks thus I suggest the publication of the manuscript in present form.